# Co-Infection of *L. monocytogenes* and *Toxoplasma gondii* in a Sheep Flock Causing Abortion and Lamb Deaths

**DOI:** 10.3390/microorganisms10081647

**Published:** 2022-08-15

**Authors:** Maria Elisabetta De Angelis, Camillo Martino, Alexandra Chiaverini, Chiara Di Pancrazio, Violeta Di Marzio, Serena Bosica, Daniela Malatesta, Stefania Salucci, Nadia Sulli, Vicdalia Aniela Acciari, Francesco Pomilio

**Affiliations:** 1Istituto Zooprofilattico Sperimentale dell’Abruzzo e del Molise “G. Caporale”, 64100 Teramo, Italy; 2Camillo Martino, Department of Veterinary Medicine, University of Perugia, Via S. Costanzo 4, 06126 Perugia, Italy

**Keywords:** *L. monocytogenes*, *T. gondii*, abortion, sheep

## Abstract

Abortion in livestock is a public health burden, and the cause of economic losses for farmers. Abortion can be multifactorial, and a deep diagnostic investigation is important to reduce the spread of zoonotic disease and public health prevention. In our study, a multidisciplinary investigation was conducted to address the cause of increased abortion and lamb mortality on a farm, which detected a co-infection of *Listeria monocytogenes* and *Toxoplasma gondii*. Hence, it was possible to conclude that this was the reason for a reduced flock health status and the cause of an increased abortion rate. Furthermore, the investigation work and identification of the *L. monocytogenes* infection root allowed the reduction of economic loss.

## 1. Introduction

Abortion in sheep flocks at a level that significantly affects productivity is a common clinical problem, potentially zoonotic. It can be multifactorial, for example, nutritional, physical, toxic, chemical, and many pathogens, such as viral, bacterial, fungal and protozoa, are frequently identified as the cause of abortions in domestic animals [1]. Data collection is fundamental to reaching an accurate diagnosis (e.g., age, location of the farm, animal movements, introduction of new animals, stage of pregnancy at the time of abortion, vaccination history, previous abortion problems, nutrition management and feeding changes, exposure to toxins or teratogenic plants, drugs administration, heat stress or stressful events such as predators on the farm, illness during pregnancy) [2].

Despite different prevalence among countries, common infectious agents of abortion in sheep are *Listeria monocytogenes* (*L. monocytogenes*), *Toxoplasma gondii* (*T. gondii*), *Campylobacter fetus* subsp. fetus (*C. fetus*), Chlamydiaceae, *Coxiella burnetii* (*C. burnetii*), *Neospora caninum* (*N. caninum*), *Salmonella abortusovis* (*S. abortusovis*), *Brucella abortus* (*B. abortus*) and *Brucella melitensis* (*B. melitensis*) [1,2]. Usually, only one etiological agent is the cause of the disease but sometimes, concurrent infection can take place [3]. Some of these pathogens are zoonotic agents that causes abortion and stillbirth not only in domestic animals but humans as well. In Europe, some monitoring for animal disease activities is in place and the results are published annually [4].

Above all, toxoplasmosis is a worldwide public health burden disease. Domestic cats and other felids are the final or definitive hosts, while humans and other animals represent intermediate hosts, which can become infected through the ingestion of oocysts from food or water contaminated by carrier cat feces or by consumption of undercooked meat containing viable tissue cysts. An infected host, in fact, can spread toxoplasma oocysts on pasture or water, which can play a role as a source of infection as well, especially in contaminated creek water. In the environment, they have been found in areas characterized by moist seasons and mild temperatures. At the correct humidity and temperature, the sporulated oocysts can survive in moist soil for up to 18 months, but their detection in feed and environmental samples is difficult due to the small number of oocysts contaminating a large volume of samples [5].

Sheep are highly susceptible to toxoplasmosis, which is a serious cause of abortion, stillbirth and economic losses. Trans-placental transmission is possible during the primary acute phase of infection in the intermediate host [6]. Not to exclude the possibility that persistent infection in ewes can lead to congenital transmission, cause infertility and abortion in sheep [7]. Fetal death and resorption are frequent during early gestation, stillborn lambs in mid-gestation and stillborn, weak or healthy lambs when the infection happens in late gestation [6].

The prevalence of the main infectious agents responsible for abortions or stillbirths in sheep and goats in different countries, and in particular in Europe, appears to be different. Among food-producing animals, T. gondii infections are more prevalent in sheep and goats than in cattle, as described by [8] Dubey and Beattie, 1988, and [9] Dubey, 2010. However, the data appear to be limited, especially in Europe. In Portugal, a seroprevalence of toxoplasmosis was estimated at 33.3% in sheep and 18.5% in goats [10]; in Romania, 53% in sheep and between 20% and 84% in goats [11]; in Spain, 46.5% in sheep and 38.3% in goats [12]; in Greece, 53.7% in sheep and 61.3% in goats [13].

In 2020, 11 Member States (MS) tested 6113 units (animals, holding or heard/flock), and 21.3% were positive for *Toxoplasma gondii* [4]. Furthermore, the seroprevalence of *T. gondii* in small ruminant meats for human consumption in Italy was reported to be 59.3% in sheep and 41.7% in goats, and less than one-year-old small ruminants showed lower seroprevalence than older animals [14].

On the other hand, listeriosis is a worldwide disease caused mainly by *L. monocytogenes* in humans and *Listeria ivanovii* in ruminants. In small ruminants, *L. monocytogenes* is the most common cause of neurological syndrome due to rhomboencephalitis, while septicemia is more likely in young lambs and abortion in the last third of gestation. The main route of transmission of the disease are feed, water or the environment contaminated by feces of sick and asymptomatic animals or discharged fetal membrane and maternal placenta of aborted animals [15]. Even more, *L. monocytogenes* and *Listeria* spp. are often detected in concomitance to higher pH and aw in silage that is incorrectly produced and/or stored. Indeed, the insufficient degree of anaerobiosis and higher pH can allow the pathogen to survive [16,17].

In 2020, in the sheep sector, 12 EU Member States (MS) tested 2015 units (animals, herd/flock and holding), and 4.5% of the units were positive for *Listeria* spp. Among them, 37 (40.7%) were reported as positive for *L. monocytogenes* [4].

Surveillance of *L. monocytogenes* in the EU is based on molecular and genotyping characterization of isolates [4]. New diagnostic tools based on whole-genome sequencing (WGS) allow deeper discrimination between isolates and enable source tracking of the disease to discover the root of the disease. Nevertheless, it can be used to improve knowledge about the ecology of *L. monocytogenes* along the food chain [4].

The aim of the paper is to describe the multidisciplinary approach used to investigate an outbreak of abortion and nervous symptoms in young lambs on a sheep farm. The investigation and laboratory tests were carried out to characterize and identify the source of the outbreak, which allowed the detection of a co-infection of multiple pathogens, above all, *T. gondii* and *L. monocytogenes*. Even more, the screening of virulence and resistance genes to antimicrobial and disinfectant was conducted for *L. monocytogenes* to investigate the specific gene profile pattern within the isolated strains.

## 2. Results

### 2.1. Necropsy and Histopathological Findings

#### 2.1.1. Description of Macroscopic Lesions

The ovine fetus and the fetal membrane with related cotyledons were edematous and partially autolytic with no exudates in the cavities. No gross lesions were observed during the fetus necropsy or inflammatory lesions in the internal organs, while the two lambs showed hyperemic meningitis and mild enteritis.

#### 2.1.2. Description of Histologic Examination

Histological examination of the organs of the fetus (kidney, brains and cotyledons) returned inflammatory lesions and the presence of protozoal cystic organisms.

Histological examination of the various organs of the lambs (kidney, liver, brain and lung) revealed, in the liver, areas of necrosis and inflammation of the portal spaces compatible with bacterial hepatitis. The other parenchyma did not show histologically detectable lesions.

Moderate suppurative meningoencephalitis with rare foci of malacia was observed. Numerous scattered colonies of small bacilli throughout the autolyzed nervous tissue were also evidenced (Figure 1).

### 2.2. Microbiological Testing

#### 2.2.1. Animal Specimen

In Table 1, all the results of the analyses carried out on the aborted fetus are summarized. *L. monocytogenes*, along with *Listeria* spp., were isolated from the liver and brain. Five colonies selected from each positive sample of liver and brain were of serogroup IVb. In Table 2, the results of MCDT carried out in the lambs are summarized; absence of bacterial growth under any condition and in all the samples was detected (brain, liver, lungs, kidney and intestine).

#### 2.2.2. Feed and Water

*Listeria* spp. was detected in four of twenty-four analytical portions tested, and more specifically, one of them was identified as *L. monocytogenes* while the other three as *L. innocua*. Five colonies collected from the samples positive for *L. monocytogenes* were of serogroup IVb. Two tested samples of water were negative for all tests. The pH values determined in the twenty-four analytical portions ranged between 3.70 and 4.40. The value of aw showed values between 0.976 and 0.995. More specifically, the four samples that tested positive for *L. monocytogenes* and *L. innocua* showed, respectively, a pH value between 3.80 and 4.20 and aw between 0.987 and 0.995 (Table 3).

### 2.3. Molecular Detection Results

#### Organ Tissues PCR

Molecular tests on the samples reported positive results for *T. gondii* in the brain of the fetus; all the results are summarized in Table 4 and Table 5.

### 2.4. Serological Testing

Out of six symptomatic ewes, four were serologically tested after about four weeks (February) of the abortion wave. They were positive for *L. monocytogenes* and *T. gondii*, and complement fixation was positive for Chlamydiacae. No antibodies were found for the other pathogens tested (*B. abortus* and *B. melitentis*, *C. brunetti*, *N. caninum* and *S. Abortusovis*). Control group ewes were not tested on this occasion.

At the second visit, in July, six months later, two of the six symptomatic ewes were not at the farm because they had been slaughtered. Among the tested ewes, three were positive for *T. gondii*, of which only two previously tested positive, one that aborted and one that delivered one small lamb. All negative results were reported for Chlamydiacae, *C. burnetii*, *L. monocytogenes* and *S. Abortusovis*. The control group instead showed 15 ewes positive for *T. gondii*, 3 to *C. burnetii*, 3 to *L. monocytogenes* and 1 serological positivity (titer 1:80) for *S. Abortusovis. N. caninum* was not tested on this occasion. The results obtained from the serological tests are shown in Appendix A.

### 2.5. Bacterial Strain L. monocytogenes

#### 2.5.1. WGS

For the 15 *L. monocytogenes* genomes analyzed, we obtained sequence data according to the quality control thresholds recommended for *L. monocytogenes* as average read quality ≥ 30, average coverage ≥ 20, de novo assembly seq. length between 2.7 and 3.2 Mbp and number of contigs ≤ 300. Furthermore, the MLST analysis confirmed that all the 15 genomes belonged to ST1 and CC1. In silico serogrouping confirmed the 4B group.

#### 2.5.2. Clustering Results

The minimum spanning tree (Figure 2) elaborated from the cgMLST analysis showed one cluster with 0–3 alleles distance. One major group of eight strains (four isolated from the brain and four from the liver fetus), a group of five strains detected in the silage and two singletons detected from the liver and brain.

#### 2.5.3. Virulence, Persistence and Antimicrobial Resistance Gene Detection Results

For virulence, 93 genes were checked; the strains shared 66 genes, and 27 were missing, among them LIPI2 and some genes encoding Internaline proteins (*inlG*, *inlL*, *inlP1*, *inlP3*, *inlP4*, *inlPq, tagB*, *ami* and *aut*) and genes of the group LM9005581 (Appendix A).

No differences were evident among the strains for SigB operon genes (lmo0889, lmo0890, lmo0891, lmo0892, lmo0893, lmo0894, lmo0895, lmo0896). The only stress island genes found were lmo1799, lmo1800 and SS1_lmo447.

Genes carrying antibiotic resistance were found in all of them, as *fosX,* lmo0919, *norB*, *sul* and, finally, *mprF* (Appendix A). No genes for metal and disinfectant resistance were detected. Moreover, the only gene in the genomic island constant in all the strains was the LGI-2_LMOSA2310; all the other genes were missing. No plasmids were detected in all of the strains.

## 3. Discussion

The present study describes an outbreak investigation following an abortion and stillbirth wave characterized by neurological symptoms in young lambs on a sheep farm. The farm, the object of our study, showed an anomalous and sudden onset of abortions and stillbirths in January 2020. Although the farmer reported abortion cases in past years, the number of lambs delivered small, dead or that showed neurological signs concerned him. Unfortunately, it was not possible to gather all the necessary information due to a lack of missing information from previous years. Wagner et al. reported the presence of the disease in the absence of evident illness in adult sheep [18].

Many agents can be the cause of abortion in a sheep flock [2]; therefore, an accurate multidisciplinary investigation was pursued to reach a correct diagnosis. Inspections of the farm were carried out at various times, and farm data were collected regarding location, flock management and nutrition, animal movements, fence-to-fence contact and biosafety measures in place, such as cleaning and disinfection, prevention of pest introduction, personnel movements and avoiding tool sharing from other farms.

Neurological symptoms, along with the detection of *L. monocytogenes* in the aborted fetus and characteristically multiple foci of necrosis through the liver and absence of macroscopical lesion in concomitance to histological findings were reported in the fetus in agreement with Broadbent [19] and consistent with other author’s observation of listeriosis encephalitic form in small ruminants [20]. Usually, septicemia is a consequence in young lambs due to bacterial dissemination and intrauterine infection of pregnant ewes [21], while the encephalitic form is more common in 4–8-month-old lambs [15]. On the other hand, Movassaghi et al. reported finding *T. gondii* in the brain of newborn lambs that died a few days after birth [22].

Histological findings of the fetus, similarly to Gual et al., also suggested toxoplasmosis infection [23]. Moreover, protozoan cysts were found in the fetus organs, and the brain tissue tested positive at PCR. The presence of *L. monocytogenes* in the same specimens examined may indicate an important role in causing abortion and neurological signs.

According to Oevermann et al., histological findings of the lambs suggested listeriosis, but *L. monocytogenes* was not confirmed through detection [24].

Concurrent infection of *L. monocytogenes* and *T. gondii* is not often reported. However, *T. gondii* is usually found in concomitance to other pathogens [25].

Noteworthily, the ewes which had abortions or stillbirths were serologically positive for *L. monocytogenes* and *T. gondii* as well. Among the sick ewes, two of them were culled, but from the remaining ewes, positivity to toxoplasmosis months later during the monitoring period was evident. This may lead us to think that the flock was persistently infected, and the ewes may have passed the infection to their offspring. This is consistent with reports by dos Santos et al. [7] and Williams et al. [26], hence the possibility of having sick, dull and neurologically impaired offspring due to the persistence of toxoplasma infection in mothers. The control group, instead, tested positive on the follow-up test. This could raise doubts about the ongoing toxoplasmosis on the farm, but knowledge of the flock’s health status at the beginning of the outbreak cannot be confirmed due to missing serological data. Indeed, blood samples could not be collected from a control group in the first instance because the farmer did not allow blood sampling for all of them, but only those showing symptoms.

Antibodies to Chlamidiacae and *L. monocytogenes* were detected in four of the investigated ewes at the first sampling, while the control group resulted as positive for *C. burnetii*, *Salmonella* Abortusovis and *L. monocytogenes* at the second blood test. Furthermore, the detection of Chlamidiacae was not confirmed at microbiological examinations since the search for Chlamidiacae in the lung by PCR was negative, so no further investigations were undertaken.

On the other hand, *L. monocytogenes* and *T. gondii* were found at serology in the sick ewes and, later, in the control group. This may have been possible due to the ongoing infection of these pathogens within the flock. Possibly, toxoplasmosis infection, once established in the flock, led to immune suppression, and the concomitant presence of other abortigenic pathogens may have determined a peak in abortion and stillbirth within the farm, causing concern for the farmer. After all, our serological results agreed with Hazlet et al. and Nayeri et al., confirming that *T. gondii* is usually found in concomitance with other pathogens and it has a higher prevalence during abortion [25,27].

In this study, the abortion wave took place between autumn and winter, confirming that *L. monocytogenes* prevalence in ruminants is higher during colder weather, probably linked to silage feeding [28]. Even more, silage feeding in the case of listeriosis is commonly reported [18,29].

The presence of *Listeria* spp. in the silage is very likely to be associated with predisposing factors allowing the growth of the bacterium. According to the EURL Listeria guideline [30], our results confirmed that the feed represented an optimal substrate for *L. monocytogenes* growth. Moreover, the use of silage in wrapped bales on the farm under study that has a greater probability of deterioration if not correctly bale, increased the risk of *L. monocytogenes* contamination, as previously reported by Fenlon and Nucera et al. [31,32].

The results obtained from the analysis of drinking water did not report positive results related to the detection of *Listeria* spp., which allowed us to exclude water as an additional source of contamination for animals. Probably because the drinking water comes from the pipeline for public drinking water, therefore, it is safe.

Overall, the health status of the flock investigated highlighted a breach of biosecurity at the farm level. Possible route causes were detected through the food and introduction of stray cats on the farm. Hence an improvement in animal and pest control was required, as well as changing the silage used as feed.

Regarding the genomic characterization, MLST results showed positive isolates belonged to the same CC1 and ST1, usually found in clinical cases, and more representative in ruminant rhomboencephalitis according to Dreyer et al. and Papić et al. [33,34]. Even more, according to Moura et al. on the base of the allelic distance, these strains belong to the same cluster [35].

The virulence profiles of the overall strains were shown to be the same and agree with Moura et al. [35]. Specifically, in our case, no ami and aut genes for resistance and invasion were detected. The inlG, inlL, inlP1, inlP3, inlP4 and inlPq belonging to the internaline gene, tagB, LIPI-4 and LIPI-2 were missing, according to Disson et al. [36].

The presence of Listeria Pathogenicity Island-1 (LIPI-1) and Listeria Pathogenicity Is-land-3 (LIPI-3) as virulence factors agreed with Cotter et al. and Disson et al. [36,37]. These pathogenicity islands, in fact, encode for Listeriolysin S (LLS), a hemolytic and cytotoxic factor necessary for *L. monocytogenes* virulence in vivo. LIPI-3 detection is concordant with Kim et al. [38], and it is mainly detected in lineage I strains. Finally, llsB is active during the systemic phase of infection [36].

SigB operon genes, as a factor for acid resistance [39], which was found in all strains, may explain their ability to resist the host gastro environment. Even more, the presence of stress island genes Lmo1800 [40], lmo1799 [41] and SSI1_lmo0447 [38] confirm the virulence of the strain caused the outbreak.

Finally, the presence of lmo0206 allows *L. monocytogenes* intracellular survival in infected macrophages, as reported in Prokop et al. [42].

No disinfectant and metal resistance were found in agreement with Disson et al. [36]. However, only LGI-2_LMOSA2310, involved in cadmium and arsenic resistance [43,44], was found in all the strains.

Antimicrobial resistance features were investigated to evaluate potential resistance to commonly used antibiotics for treatment; the cationic antimicrobials peptides (CAMP) gene mprF was found in all strains, according to Zuber et al. [45]. Even more, the resistance gene to fosfomycin (fosX), lincosamides (lmo0919), quinolones (norB) and sul-phonamides (sul) was found, as expected [46].

## 4. Conclusions

In this study, we presented a multidisciplinary approach for an abortion outbreak that is fundamental to reaching a correct diagnosis.

Our results showed that co-infection of *L. monocytogenes* and *T. gondii* was the most probable cause of abortion wave in this sheep flock, and feed played a key role as a vector of infection.

At the same time, *L. monocytogenes* was found in the fetus and feed with the concomitant presence of protozoal cysts in fetus organs. Possibly, the infection of toxoplasmosis led to a flock immune depression, which caused an abortion peak noticed by the farmer only later during the lambing season. This hypothesis is supported by PCR positive results and is consistent with the serological positivity of *T. gondii* and serological conversion after the abortion wave to *L. monocytogenes*.

Unfortunately, during the investigation, it was difficult to collect all the sick and dead animals. To increase the evidence of the results is of paramount importance to collect as many sick animals as possible. Even more, during a farm investigation, quite often, it is difficult to collect reliable information on the history of the farm if the data collection is not in place.

Despite this, a deep investigation work in the case of abortion outbreaks and source tracking is important to contain animal and economic loss. As in our case, the elimination of the contaminated silage from the diet was a good measure of prevention to stop the diffusion of the abortion wave, although the consequence may have a big economic impact on the farm due to an increase in costs if more feed has to be bought.

Biosecurity measures to prevent the introduction of disease were reported as in place. Despite this, the possible breach of security measures may have determined the abortion outbreak, especially in the case of *T. gondii*.

Finally, the clustering analysis showed that positive samples from the fetus shared a similar allelic profile to the five samples from the silage. Hence it identified the feed matrix as a source of infection. Virulence investigation of the isolated bacteria strengthened our knowledge of the pathogen characteristics, and antibacterial resistance genes screening is a good habit for investigating potentially harmful strains for prevention purposes.

## 5. Materials and Methods

### 5.1. Farm Data

The sheep farm that was the object of this study is located in the province of L’Aquila (Abruzzo, Italy). In January 2020, out of 21 ewes, a total of 3 ewes delivered small lambs, while 1 ewe aborted. Later on, during the lambing season, 2 more ewes gave birth to lambs that died a few days later with neurological symptoms. One of the aborted fetuses and two more lambs were sampled and sent to the laboratories of Istituto Zooprofilattico Sperimentale of Abruzzo and Molise, Teramo, Italy (IZSAM) for necropsy and microbiological investigations. The rest of the flock did not show any clinical symptoms. In previous years on the farm, there had never been such a high number of abortions. However, exact figures were not available.

To carry out an epidemiological investigation, the farm was visited three times: in January 2020 at the onset of the abortion wave, after four weeks and six months later.

The entire sheep population was evaluated as a closed flock; all the farmed animals lived in a single group sharing the same spaces and biosecurity measures were in place to prevent direct contact with outside animals. The flock was housed in a single enclosure with access to the pasture during summer.

During the summer, the flock was fed grass and forage; in winter, hay and grass silage produced on the farm and stored as silage bales were used as the feed. The drinking water for the animals was supplied from the public water pipeline. The silage used before the epidemic was not available for analysis. During the visit to the farm, no traces of contamination or color and structure alterations were found in the remaining bales, which were considered of good quality. Nevertheless, 12 random samples from the silage bales were collected along with 2 samples of drinking water from the tank and microbiology analysis, pH and water activity (aw) [47] were carried out. In addition, blood samples from both symptomatic and asymptomatic animals were taken during the second and the third visit for a serological survey of abortion disease.

### 5.2. Necroscopy and Histological Examination

The necropsy of fetus and lambs was performed, and brain, liver, lungs, spleen and cotyledons samples from fetus, and brain, lungs, livers, small intestines, caecal contents, kidneys and spleens from lambs were collected for microbiological and molecular investigations.

Moreover, fetal kidney, brain and cotyledons, and lamb kidneys, livers, brains and lungs underwent histological examination.

### 5.3. Microbiological Testing

Fetal brain, liver lungs and cotyledons were tested for the detection of abortive pathogens (*Brucella* spp., *Listeria* spp., *L. monocytogenes*, *Salmonella* spp., *Campylobacter* spp.), and lamb brains, lungs, livers, small intestines, cecal content and kidneys were tested for aerobically and anaerobically microbiological culture-dependent test (MCDT), *L. monocytogenes* and *Salmonella* spp.

Aerobically and anaerobically MCDT on the aborted fetus and lamb organs were performed as per the internal procedure for abortive detection at IZSAM, based on analytical methods as described by Koneman et al., (2019) [48], Krieg et al., (2005) [49], Quinn et al., (2013) [50] and Jouseimies-Somer et al., (2002) [51].

The detection of Listeria spp. and *L. monocytogenes* was performed on 24 analytical portions from the 12 silage bails samples and 2 water samples according to ISO 11290-1: 2017 [52].

### 5.4. Molecular Detection

Fetal brain, liver, lungs, spleen and cotyledons were collected for the detection of *Brucella* spp., toxoplasmosis, *Neospora* spp., Schmallenberg virus (SBV), *Leptospira* spp., Chlamydiaceae, Border Virus Disease (BVD) and Bluetongue Virus (BTV), and lamb brain, kidney and spleen for the detection of *T. gondii*, *Neospora* spp. and BTV.

Molecular detection of abortive agents (*Brucella* spp, *T. gondii*, *N. caninum*, Schmallenberg virus, *Leptospira* spp. Chlamydiacae, BDV and BTV) was carried out on the samples taken from the fetus and two lambs with neurological symptoms. DNA extraction was executed with a Maxwell^®^ 16 Cell and Tissue DNA Purification Kit (Promega Italia Srl, Milan, Italy). The DNA purity was checked by NanoDrop2000 (ThermoFisher Scientific, Waltham, MA, USA). The mix was prepared using the Genesig Advanced Kit (Genesig, York House, School Lane, Chandler’s Ford, UK), according to the manufacturer’s instructions, and Real-Time PCR was performed (QuantStudio™ 7 Pro Applied Biosystems, CA, USA).

### 5.5. Serological Testing

After the detection of *L. monocytogenes* in aborted fetuses, blood samples were taken four weeks and six months after the abortion from symptomatic ewes, mothers of dead or sick lambs. No control group was sampled at the first sampling for a total of four sick animals, while for the second blood sampling, 16 ewes were chosen as the control group plus 2 more symptomatic ewes, for a total of 22 sheep to be evaluated for the detection of antibody title of the IgG anti-*L. monocytogenes* by “Listeria sheep Test” ELISA kit (DIATHEVA, Italy), Chlamydiaceae by Complement Fixation Test (CTF), *C. burnetii* by ID Screen Q Fever Indirect Multi-species (IDvet, France), *N. caninum* (ID Screen *Neospora caninum* Competition, IDvet, France), *S*. Abortusovis by serum agglutination test and *T. gondii* by Indirect fluorescent antibody (IFA) test. The serological methods are reported in the Appendix A.

### 5.6. Listeria Monocytogenes Molecular Testing

#### 5.6.1. DNA Extraction

Five suspected colonies were selected for typing from each positive sample of fetus brain, liver and silage (15 colonies in total), and later sent to the National Reference Laboratory for *L. monocytogenes* (NRL*Lm*) of IZSAM for whole-genome sequencing (WGS).

DNA of *L. monocytogenes* strains was extracted using a Maxwell^®^ 16 tissue DNA purification kit (Promega Italia Srl, Milan, Italy) according to the manufacturer’s protocol, and the DNA purity was checked by NanoDrop2000 (ThermoFisher Scientific, Waltham, MA, USA).

#### 5.6.2. Molecular Serogrouping

Molecular serogrouping for *L. monocytogenes* was performed for all the strains using a multiplex PCR assay based on the amplification of the same targets as described by Doumith et al. [53] and Kérouanton et al. [54]: prs, lmo0737, ORF2110, lmo1118, ORF2819 and the Lm-specific gene prfA.

#### 5.6.3. Next-Generation Sequencing (WGS) and Data Analysis

Starting from 1 ng of input DNA, the Nextera^®^ DNA Library Preparation kit was used for library preparation according to the manufacturer’s protocols. WGS was performed on NextSeq^®^ 500 platform (Illumina^®^, San Diego, CA, USA) with the NextSeq 500/550 mid-output reagent cartridge v2 (300 cycles, standard 150-bp paired-end reads) according to the manufacturer’s instructions.

For the WGS data analysis, an in-house pipeline [55] was used. The trimming step of raw reads was performed using Trimmomatic [56], and a quality control check of the reads using FastQC v.0.11.5., which are included in the de novo assembly of paired-end reads, was carried out using SPAdes v3.11 [57] with default parameters for the Illumina platform 2 × 150 chemistry. Finally, the quality check of the genome assemblies was performed with QUAST v.4.3.

All the genomes that met the quality parameters recommended by Timme et al. [58] were used for the subsequent analysis steps. The MLST scheme, based on seven housekeeping genes, was used to characterize *L. monocytogenes* strain sequence type (ST) and Clonal complex (CC) querying the BIGSdb-Lm platform (https://bigsdb.pasteur.fr/listeria/).

#### 5.6.4. Cluster Analysis

The core genome MLST (cgMLST) of *L. monocytogenes* was calculated according to the Institute Pasteur’s scheme of 1748 target loci using the chewBBACA allele calling algorithm [59]. Genomes with at least 1660 called loci (95% of the full scheme) were considered. A Minimum Spanning Tree was visualized using GrapeTree [60].

#### 5.6.5. Virulence Genes and Genetic Determinant Involved in Persistence and Antibiotic Resistance

Genome assemblies were manually screened for the presence/absence of virulence genes, loci encoding for disinfectants, metal and antimicrobial resistance genes, using different functions available on BIGSdb-Lm platform (http://bigsdb.pasteur.fr/listeria).

The heatmap of the virulence and antibiotic resistance genes was elaborated by online available software Morpheus (https://software.broadinstitute.org/morpheus). Plasmid replicons were checked using PlasmidFinder tool (https://cge.cbs.dtu.dk/services/PlasmidFinder).

## Figures and Tables

**Figure 1 microorganisms-10-01647-f001:**
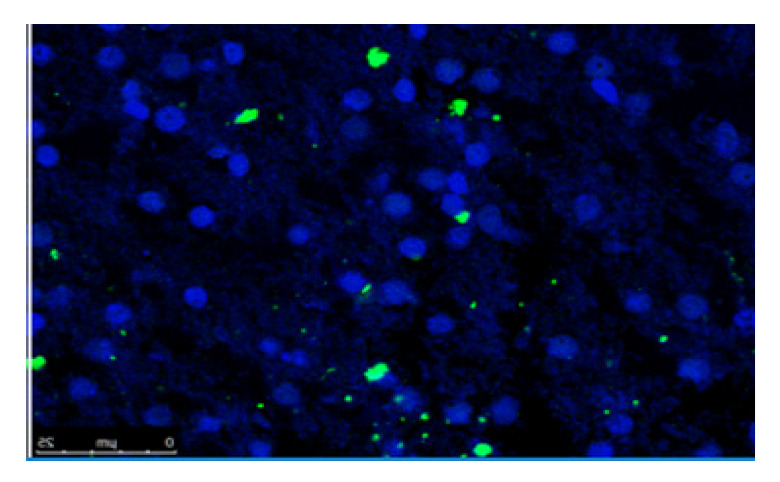
Brainstem section. In Situ detection of *Listeria monocytogenes* antigen (green color) and cellular nuclei (blue color), scale bar: 25 µm.

**Figure 2 microorganisms-10-01647-f002:**
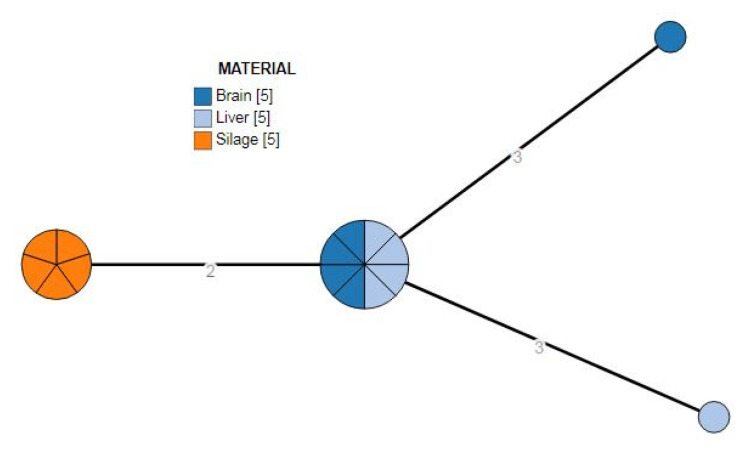
cgMLST of the 15 *L. monocytogenes* strains.

**Table 1 microorganisms-10-01647-t001:** Results of microbiological culture-dependent testing on aborted fetus. ^Ø = Not Tested.^

	*Brain*	*Liver*	*Lungs*	*Cotyledons*
*Brucella* spp.	Negative	Negative	Negative	Negative
*Listeria* spp.	Positive	Positive	Ø	Ø
*L. monocytogenes*	Positive	Positive	Ø	Ø
*Salmonella* spp.	Ø	Negative	Ø	Ø
*Campylobacter* spp.	Ø	Negative	Ø	Ø

**Table 2 microorganisms-10-01647-t002:** Results of microbiological culture-dependent testing on two lambs. ^Ø = Not Tested.^

	MCDT Aerobically	MCDT Anaerobically	*L. monocytogenes* Detection	*Salmonella* spp. Detection
Brain Lamb 1	Negative	Negative	Negative	Ø
Lung Lamb 1	Negative	Negative	Ø	Ø
Liver Lamb 1	Negative	Negative	Negative	Negative
Small Intestine Lamb 1	Negative	Negative	Ø	Ø
Cecal Lamb 1	Negative	Negative	Ø	Ø
Kidney Lamb 1	Negative	Negative	Negative	Ø
Brain Lamb 2	Negative	Negative	Negative	Ø
Lung Lamb 2	Negative	Negative	Ø	Ø
Liver Lamb 2	Negative	Negative	Negative	Negative
Small Intestine Lamb 2	Negative	Negative	Ø	Ø
Cecal Lamb 2	Negative	Negative	Ø	Ø
Kidney Lamb 2	Negative	Negative	Negative	Ø

**Table 3 microorganisms-10-01647-t003:** *Listeria* spp. and *L. monocytogenes* culture-dependent testing on feed and water. ^Ø = Not tested.^

Material Tested	Samples Tested (N.)	Analytical Portions Tested (N.)	*L. innocua* Detected	*L. monocytogenes* Detected	pH	aw
Silage grass (25 g)	12	24	3	1	3.957 (3.7–4.4)	0.983 (0.976–0.995)
Drinking water (500 mL)	2	2	0	0	Ø	Ø

**Table 4 microorganisms-10-01647-t004:** Results of PCR testing of fetus. ^Ø = Not tested.^

	*Brain*	*Liver*	*Lungs*	*Spleen*	*Cotyledons*
*Brucella* spp.	Negative	Negative	Negative	Ø	Negative
*T. gondii*	Positive	Ø	Ø	Ø	Ø
*N. caninum*	Negative	Ø	Ø	Ø	Ø
Schmallenberg Virus	Negative	Ø	Ø	Ø	Ø
*Leptospira* spp.	Ø	Ø	Negative	Ø	Ø
*Chlamydiaceae*	Ø	Ø	Negative	Ø	Ø
Border Disease Virus	Ø	Ø	Ø	Negative	Ø
Bluetongue Virus	Ø	Ø	Ø	Negative	Ø

**Table 5 microorganisms-10-01647-t005:** Results of molecular testing of two lambs. ^Ø = Not tested.^

	*Brain*	*Kidney*	*Spleen*
*T. gondii*	Negative	Ø	Ø
*N. caninum*	Negative	Negative	Ø
Bluetongue Virus	Ø	Ø	Negative

## Data Availability

At the moment of submission, the authors will submit the genomic sequencing to NCBI as soon as possible.

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
