# Peer review of "Co-Infection of L. monocytogenes and Toxoplasma gondii in a Sheep Flock Causing Abortion and Lamb Deaths"

_microorganisms, 2022, doi:10.3390/microorganisms10081647_

Round 1

Reviewer 1 Report

General comments

The paper subject is interesting, from an epidemiological point of view, and particularly in the pathology of small ruminants. Some doubts about methodology and epidemiology deserve to be clarified. A review of English is suggested, allowing for robust scientific writing that values the text.

Specific comments

Line 15 - The investigation work and source tracking allowed understanding the health status of the flock and reducing economic loss.

The investigation is more a contribution to the knowledge of the health status and measures to reduce economic losses were not described.

Line 30 - Despite different prevalence among countries, the most common infectious agents of abortion in sheep are

In addition to not providing scientific support (references) would be interesting to develop information on the prevalence of these pathologies in different countries, particularly in Europe. In addition, regarding Listeriosis and Toxoplasmosis, relevant epidemiological information should be provided, namely prevalence values (particularly in ruminants).

Line 67 Aim of the paper is to describe an outbreak o

Is the purpose of describing an outbreak? Thus, you will have to have much more of clinical signs, treatment, prophylaxis and other elements that are decisive in the analysis of the outbreak.

Table - Ø= Not Tested (as legend)

Line 90 - Microbiological testing

Why wasn't research done on Chlamydiaceae considering its epidemiological and health impact on the herd?

Line 116 - Serological test

The screening procedure between groups and moments needs strong justification, given the low coherence of the sampling.

Line 166 - The farm, object of our study, showed an anomalous and sudden onset of abortions and stillbirths in January 2020.

The information provided in the material and methods is scarce regarding the incidence and prevalence of abortions. There are 2 to 3 abortions in how many sheep. Why is it considered abnormal? There are no abortions in previous years?

The farmer reported abortion cases before, but never so many (2-3?)

Line 266 - Biosecurity measures to prevent introduction of disease were reported as in place. Despite this, possible breach of security measures may have determined the abortion outbreak, especially in the case of T. gondii.

This is an important point – biosafety – that was not addressed in the discussion. Neither the existing (and failed) measures nor the reinforcement ones.

277 – A better characterization of Farm data would be positive, namely in terms of flock.

284 - there had never been such a high number of abortions (3?). However, exact figures were not available (I think it is difficult to understand the lack of knowledge of the number of abortions, being reduced - better clarification)

In general, interesting subject but should be enriched by small corrections and information to be published.

Reviewer 2 Report

The manuscript by Angelis et al. describe a full analysis for the identification of the cause of an outbreak within a sheep farm, leading to abortion and nervous symptoms. The study showed a complete analysis and the results are important to the field, however the manuscript showed a significant lack of information, mostly in the methodology section. The scientific terminology also need to be improve, as well as the English writing.

I recommended major modification, before its publication. The comments raised below would help the authors to revise the manuscript and clarify some of the information.

The Introduction need to be re-formulate, as is focus in the results obtained.  Detailed information about L. monocytogenes and T. gondii that should be presented in the discussion to support the results. In this section, the authors may include more information about past outbreak or general information about the causes, not only focus in the other two pathogens.

Need to be explained why the authors performed the NGS and the detection of antimicrobial resistance genes, as it was not mentioned before. I suggest to also include something about this topic in the introduction.

The methodology need to be revised and improve as information is missing:

·         How was performed DNA extraction, which primers were used and why this target genes.

·         What technique/ equipment was used for the NGS and the protocol followed

·         The way how the DNA extraction was performed and primers used for the PCR analysis.

·         Also modify the titles of the sections mentioning: “Detection of Antibodies Vs…” as you are not detecting the antibodies, but the antigens of the pathogens. Suggestion: Immunological detection of….”

Round 2

Reviewer 1 Report

The changes made by the authors to this paper article are positive and allow a better understanding of their objectives and methodology. However, both the sample and the epidemiological study (not carried out) are important scientific weaknesses. The justification of non-authorization by the producer, even if valid, disturbs the experimental design and its scientific value.

Reviewer 2 Report

Dear Authors,

All modifications were correctly performed.

Author Response

Following the positive considerations of the reviewer, no changes were made.